# Pruning deep neural networks generates a sparse, bio-inspired nonlinear controller for insect flight

**Olivia Zahn** [1]*, **Jorge Bustamante Jr.** [2], **Callin Switzer** [2], **Thomas L. Daniel** [2], **J. Nathan Kutz** [3,4]

**1** Department of Physics, University of Washington, Seattle, Washington, United States of America,
**2** Department of Biology, University of Washington, Seattle, Washington, United States of America,
**3** Department of Applied Mathematics, University of Washington, Seattle, Washington, United States of America, **4** Department of Electrical and Computer Engineering, University of Washington, Seattle, Washington, United States of America

\* otthomas@uw.edu

**Data Availability Statement:** All code and data associated with the simulations and the DNNs is available on Github at https://github.com/oliviatessa/MothPruning.

## Abstract

Insect flight is a strongly nonlinear and actuated dynamical system. As such, strategies for understanding its control have typically relied on either model-based methods or linearizations thereof. Here we develop a framework that combines model predictive control on an established flight dynamics model and *deep neural networks* (DNN) to create an efficient method for solving the inverse problem of flight control. We turn to natural systems for inspiration since they inherently demonstrate network pruning with the consequence of yielding more efficient networks for a specific set of tasks. This bio-inspired approach allows us to leverage network pruning to optimally sparsify a DNN architecture in order to perform flight tasks with as few neural connections as possible, however, there are limits to sparsification. Specifically, as the number of connections falls below a critical threshold, flight performance drops considerably. We develop sparsification paradigms and explore their limits for control tasks. Monte Carlo simulations also quantify the statistical distribution of network weights during pruning given initial random weights of the DNNs. We demonstrate that on average, the network can be pruned to retain a small amount of original network weights and still perform comparably to its fully-connected counterpart. The relative number of remaining weights, however, is highly dependent on the initial architecture and size of the network. Overall, this work shows that sparsely connected DNNs are capable of predicting the forces required to follow flight trajectories. Additionally, sparsification has sharp performance limits.

## Author summary

Originally inspired by biological nervous systems, deep neural networks (DNNs) are powerful computational tools for modeling complex systems. DNNs are used in a diversity of domains and have helped solve some of the most intractable problems in physics, biology,

**Funding:** This work was supported by AFOSR Grant (Nature-Inspired Flight Technology and Ideas, FA9550-14-1-0398) to TLD and the Komen Endowed Chair to TD. CS was supported in part by the Washington Research Foundation and by a Data Science Environments project award from the Gordon and Betty Moore Foundation (Award \#2013-10-29) and the Alfred P. Sloan Foundation (Award \#3835) to the University of Washington eScience Institute. JB was supported in part by the National Science Foundation Graduate Research Fellowship Program. JNK and TD acknowledge support from the Air Force Office of Scientific Research (FA9550-19-1-0386). The funders had no role in study design, data collection and analysis, decision to publish, or preparation of the manuscript. Nature-Inspired Flight Technology and Ideas: http://washington.edu/#:~:text=Technologies%20and%20Ideas-,The%20Air%20Force%20Center%20of%20Excellence%20on%20Nature%2DInspired%20Flight,of%20flight%20in%20complex%20environments. Washington Research Foundation: https://www.wrfseattle.org/ Gordon and Betty Moore Foundation: https://www.moore.org/ Alfred P. Sloan Foundation: https://sloan.org/ NSF: https://www.nsf.gov/ Air Force Office of Scientific Research: https://www.afrl.af.mil/AFOSR/.

**Competing interests:** The authors have declared that no competing interests exist.

and computer science. Despite their prevalence, the use of DNNs as a modeling tool comes with some major downsides. DNNs are highly over-parameterized, which often results in them being difficult to generalize and interpret, as well as being incredibly computationally expensive. Unlike DNNs, which are often trained until they reach the highest accuracy possible, biological networks have to balance performance with robustness to a noisy and dynamic environment. Biological neural systems use a variety of mechanisms to promote specialized and efficient pathways capable of performing complex tasks in the presence of noise. One such mechanism, synaptic pruning, plays a significant role in refining task-specific behaviors. Synaptic pruning results in a more sparsely connected network that can still perform complex cognitive and motor tasks. Here, we draw inspiration from biology and use DNNs and the method of neural network pruning to find a sparse computational model for controlling a biological motor task.

## Introduction

Between childhood and adolescence, the number of synaptic connections between neurons sharply decreases through a process called synaptic pruning [1]. Depending on the neural system, synaptic pruning can improve the brain's efficiency and affect cognitive function. In fact, synaptic pruning is seen as a mechanism for learning, in which the environment affects which neural connections are maintained and which are removed [2]. Refinement of neural connections via pruning occurs in wide ranging taxa, from humans to *Drosophila* and in systems ranging from sensory input to motor control [3, 4]. For example, during metamorphosis of the hawkmoth, *Manduca Sexta*, synapses are pruned and reconnected to enable adult-specific behaviors [5]. The biological mechanisms that underlie synaptic pruning (often activity dependent) have a range of processes including a variety of semaphorins, increased GABAergic signaling, changes in dendritic spine density (with some enigmatic mechanisms), and even neuro-immune interactions [6]. Synaptic pruning plays a significant role in refining task-specific behaviors, the result of which is a more sparsely connected network that can still perform complex cognitive and motor tasks.

There is a rich basis of literature in biological synaptic pruning. However, the main finding across these numerous studies is that synaptic pruning plays a major role in the refinement of neural connectivity [3]. Through the overgrowth of synapses and their subsequent pruning, biological neural systems are made both optimal for a specific task and more efficient for having more sparse connectivity. Deep neural networks (DNNs), which were originally motivated by the visual cortex of cats and the pioneering work of Hubel and Wiesel [7, 8], are often considered as mathematical proxies for biological neural processing. The universal approximation properties of DNNs [9] make them ideal for modeling high-dimensional, complex, nonlinear mappings for a large diversity of problems. From image and speech recognition [10, 11] to fluid flow control [12, 13], DNNs learn input-to-output mappings by combining gradient descent with the backpropagation algorithm. Like biological pruning, DNNs have an extensive literature dedicated to improving the generalization capabilities (i.e. performance on unseen data) and computational efficiency of DNNs through the mechanism of pruning.

The sparsification of such DNNs has typically been motivated by the pernicious effects of over-fitting data, and to a lesser extent, the DNNs computational and memory footprint, i.e. the need to be implemented on small portable devices such as smart phones. Dropout, for instance, was one of the early versions of sparsification that allowed for greater generalization capabilities [11, 14, 15]. However, standard dropout methods typically only enforce *temporary*

sparsification since the algorithm often allows for nodes to again re-train their weights from their zeroed-out state. Thus most DNNs typically remain highly over-parameterized and their layers are fully-connected. For example, the natural language processing model, GPT-3, is the largest DNN ever built with 175 billion parameters [16], and successful models with millions of parameters are not uncommon. There are many different methods to make DNNs more sparse, ranging from regularization during training [17] to specifying sparse architectures [18].

Biologically inspired neural network pruning has also been shown to be an effective method for sparsifying a DNN without compromising performance [19–23]. In neural network pruning, the connectivity of a DNN is made more sparse by forcing select weights between the layers to zero and then retraining, resulting in a more sparse network that is capable of performing comparably to the fully-connected network up to a certain limit. Pruning has been used to prevent network overfitting and to reduce overall model size. Pruned DNNs have the advantage of (i) having a small memory footprint, (ii) providing improved generalization, and (iii) being more efficient for generating input-output computations. Thus they have important practical advantages over their fully-connected counterparts. They are also more representative of biological neural systems, in which neural pathways are sparsely and specifically connected for task performance. In fact, a diversity of sparse networks exist across species. For example, the respiratory rhythm patterns of mammals are generated by sparsely connected networks [24]. In the olfactory system of *Drosophila*, high-dimensional odor signals are sparsely encoded via the mushroom body [25, 26]. Neural network pruning enables the exploration of biologically-inspired, sparse learning and the strengths of the resultant sparse networks.

The inverse problem of insect flight is a highly nonlinear dynamical system, in part due to the unsteady mechanisms of flapping flight [27, 28] and the noisy environment through which insects maneuver. As such, the inverse problem of insect flight serves as an exemplar to study whether a DNN can solve a biological motion control problem while maintaining a sparse connectivity pattern. In an inverse problem, the initial and final conditions of a dynamical system are known and used to find the parameters necessary to control the system. In other words, the DNN in this study is trained to predict the controls required to move the simulated insect from one state space to another. Solving the inverse problem of insect flight has been previously simulated using a genetic algorithm wedded with a simplex optimizer for hawkmoth level forward flight and hovering [29]. Another study linearized the dynamical system of simulated hawkmoth flight and found the system to operate on the edge of stability [30]. Recently, a study developed an inertial dynamics model of *M. sexta* flight as it tracked a vertically oscillating signal, which modeled the control inputs using Monte Carlo methods in a model-inspired by model predictive control (MPC) [31].

In this work, we use the inertial dynamics model in [31] to simulate examples of *M. sexta* hovering flight. Fig 1 shows the physical parameters of the simulated moth and the inertial dynamics model. These data are used to train a DNN to learn the controllers for hovering. Drawing inspiration from pruning in biological neural systems, we sparsify the network using neural network pruning. Here, we prune weights based simply on their magnitudes, removing those weights closest to zero. Importantly, the pruned weights remain zeroed out throughout the sparsification process. This bio-inspired approach to sparsity allows us to find the optimally sparse network for completing flight tasks. Insects must maneuver through high noise environments to accomplish controlled flight. It is often assumed that there is a trade-off between perfect flight control and robustness to noise and that the sensory data may be limited by the signal-to-noise ratio. Thus the network need not train for the most accurate model since in practice noise prevents high-fidelity models from exhibiting their underlying

## (A) Schematic of simulated moth

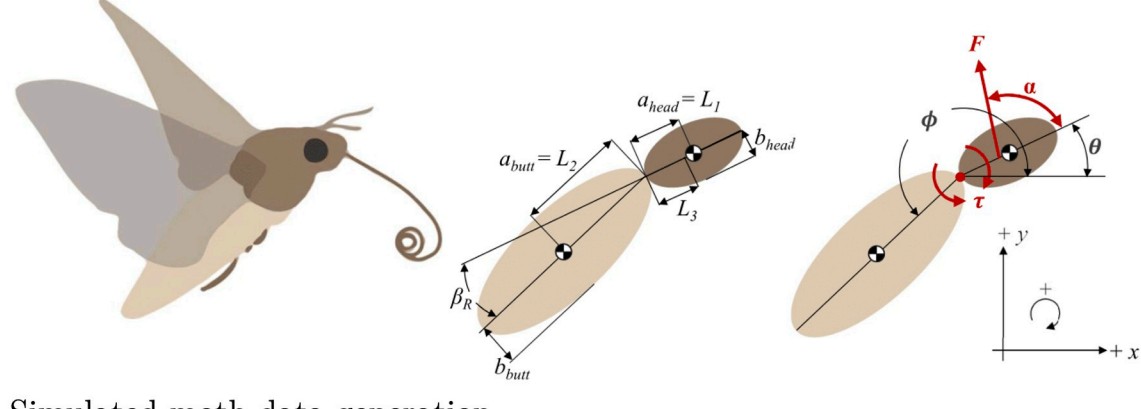

## (B) Simulated moth data generation

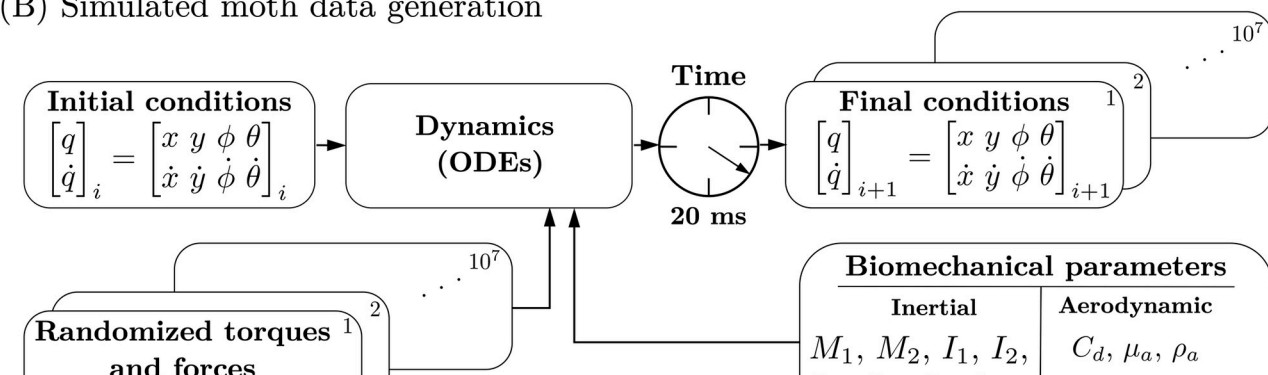

## (C) Neural network inverse model

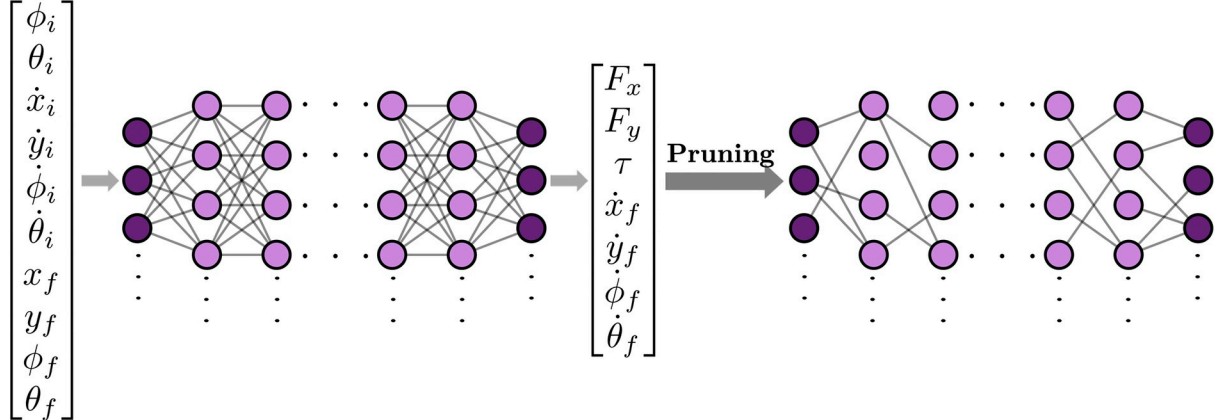

**Fig 1. Inverse problem of flight control.** (A) The moth body is made of two ellipses attached with a spring. There are three control variables ($F$, $\alpha$, and $\tau$) and four parameters to describe the state space ($x$, $y$, $\theta$, and $\phi$). See S1 Table for the global parameters and S2 Table for the calculated variables. (B) The differential equation solver solves the forward problem of insect flight control. (C) The neural network is an attempt to solve the inverse problem of flight control.

accuracy. Rather, we seek to find the sparsest model capable of performing the task given the noisy environment. We employed two methods for neural network pruning: either through manually setting weights to zero or by utilizing binary masking layers. Furthermore, the DNN is pruned sequentially, meaning groups of weights are removed slowly from the network, with retraining in-between successive prunes, until a target sparsity is reached. Monte Carlo simulations are also used to quantify the statistical distribution of network weights during pruning given random initialization of network weights. This work shows that sparse DNNs are capable of predicting the controls required for a simulated hawkmoth to move from one state-space to another, or through a sequence of control actions. Specifically, for a given signal-to-noise level the pruned network can perform at the level of the fully-connected network while requiring only a fraction of the memory footprint and computational power.

## Results

### Network pruning results

Fig 2 shows the learning curve for a network trained using the sequential pruning protocol with TensorFlow's Model Optimization Toolkit (see Methods section for details) [32]. The network is trained until a minimum error is reached, and then pruned to a specified sparsity percentage and then retrained until the loss is once again minimized. The sparsity (or pruning) percentages are shown in Fig 2 where they occur in the training process. An arbitrary threshold error of $10^{-3}$ (shown as a red, dashed line) was chosen to define the optimally sparse network (i.e. sparsest possible network that performs under the specified loss). This specific threshold value was chosen because it is near the performance of the trained, fully-connected network. In practice, the red line represents the noise level encountered in the flight system.

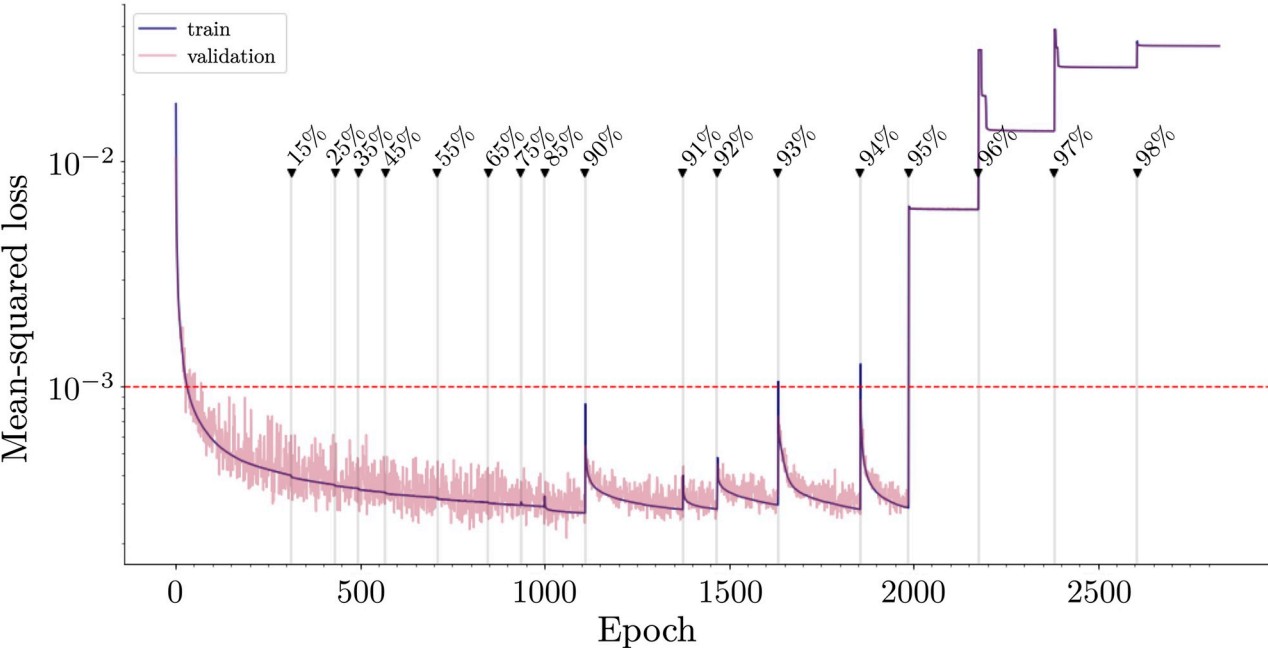

**Fig 2. Learning curve for sequential pruning of network.** Fully-connected neural network is trained until the mean-squared error is minimized. Then, the network is sequentially pruned by adding in masking layers and trained again. The performance of the network improves below the minimum error achieved by the fully-connected network for low levels of pruning, but performs comparably to the fully-connected network until 94% of the network is pruned.

Specifically, given a prescribed signal-to-noise ratio, we wish to train a DNN to accomplish a task with a certain accuracy that is limited by noise. Thus high-fidelity models, which can only practically exist with perfect data, are traded for sparse models which are capable of performing at the same level for a given noise figure. In the example in Fig 2, the optimally sparse network occurs at 94% sparsity (or when only 6% of the connections remain). Beyond 94% sparsity, the performance of the network breaks down because too many critical weights have been removed.

## Monte Carlo results

To compare the effects of pruning across networks, we trained and pruned 1320 networks with different random initializations on the same dataset. In this experiment, the hyper-parameters, pruning percentages, and architecture are held constant. Fig 3 shows the training curves of 9 sample networks. The red, dashed line in each of the panels represents the same threshold as in Fig 2 ($10^{-3}$). The black, solid lines in Fig 3 represent the optimally sparse networks. Although the majority of networks in this subset breakdown at 93% sparsity, a few breakdown at higher and lower levels of connectivity.

Fig 4 shows the loss after pruning the 1320 networks at varying pruning percentages (from 0% sparsity to 98% sparsity). The box plot in Fig 4 is directly comparable to Fig 2, but it is the compilation of the results for many different networks. The networks do not all converge to the same set of weights, which is evident by the numerous outliers, as well as the variance around the median loss.

The median minimum loss achieved by the networks before pruning is 7.9 x $10^{-4}$. The first box in the box plot in Fig 4 corresponds to the losses of all the trained networks before any pruning occurs. The variance on the loss is relatively small, but there are several outliers. Once again, the red, dashed line in the box plot in Fig 4 represents the threshold below which a network is optimally sparse. Many networks follow a similar pattern and perform under the threshold until they exceed 93% sparsity. Also, many networks perform better than the median performance of the fully-connected networks when pruned up to 85% sparsity.

The number of optimally sparse networks in each sparsity category is shown in the bar plot at the top of Fig 4. Of the 1320 networks trained, 858 of the networks are optimally sparse at 93% sparsity. A small number of networks (5) remain under the threshold up to 95% pruned. Note that the total number of networks represented in the bar plot does not add up to 1320. This is because several networks never perform below the threshold throughout the sequential pruning process (see outliers in Fig 4).

**Analysis of layer sparsity.** The subset of optimally sparse networks pruned to 93% is used in the following analysis of network structure (858 networks). The sparsity across all the layers was found to be uniform (7% of weights remain in each layer) despite not explicitly requiring uniform pruning in the protocol. Table 1 shows the average number of remaining connections across the 858 networks, as well as the variance and the fraction of remaining connections.

Fig 5 shows a box plot of the number of connections from the input layer to the first hidden layer for the subset of pruned networks. Interestingly, the initial head-thorax angular velocity was completely pruned out of all of the networks in the subset, meaning it has no impact on the output and predictive power of the network. Additionally, the initial abdomen angular velocity connects to either zero, one, or two nodes in the second hidden layer, while all the other inputs have a median connection to at least 5% of the weights in the first hidden layer.

## Materials and methods

All code associated with the simulations and the DNNs is available on Github [33].

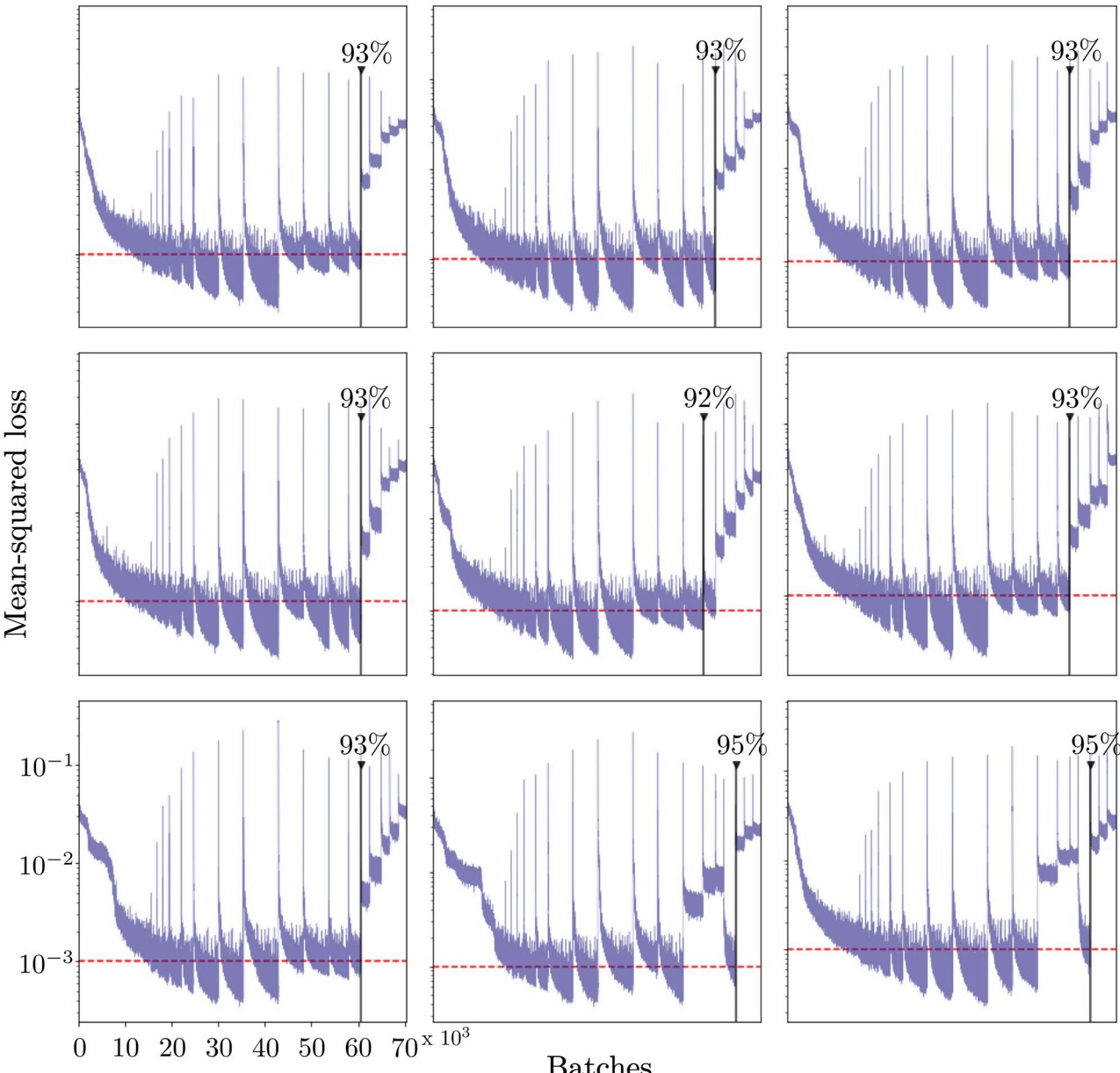

**Fig 3. Performance breakdown of 9 sample pruned networks.** The networks are sequentially pruned. Each network is evaluated to find the optimally sparse network. The red dashed line represents the performance threshold ($10^{-3}$). The sparsest network that performs below this threshold is shown by the solid, black vertical line.

## Moth model

The simulated insect uses an inertial dynamics model developed in Bustamante et al., 2021 [31] and was inspired by hawkmoth flight control, *M. sexta* with body proportions rounded to the nearest 0.1 cm. The simulated moth was made up of two ellipsoid body segments, the head-thorax mass ($m_1$) and the abdomen mass ($m_2$). The body segments are connected by a pin joint consisting of a torsional spring and a torsional damper as seen in [34]. The simulated moth model could translate in *x-y* plane, and both the head-thorax mass, and the abdominal

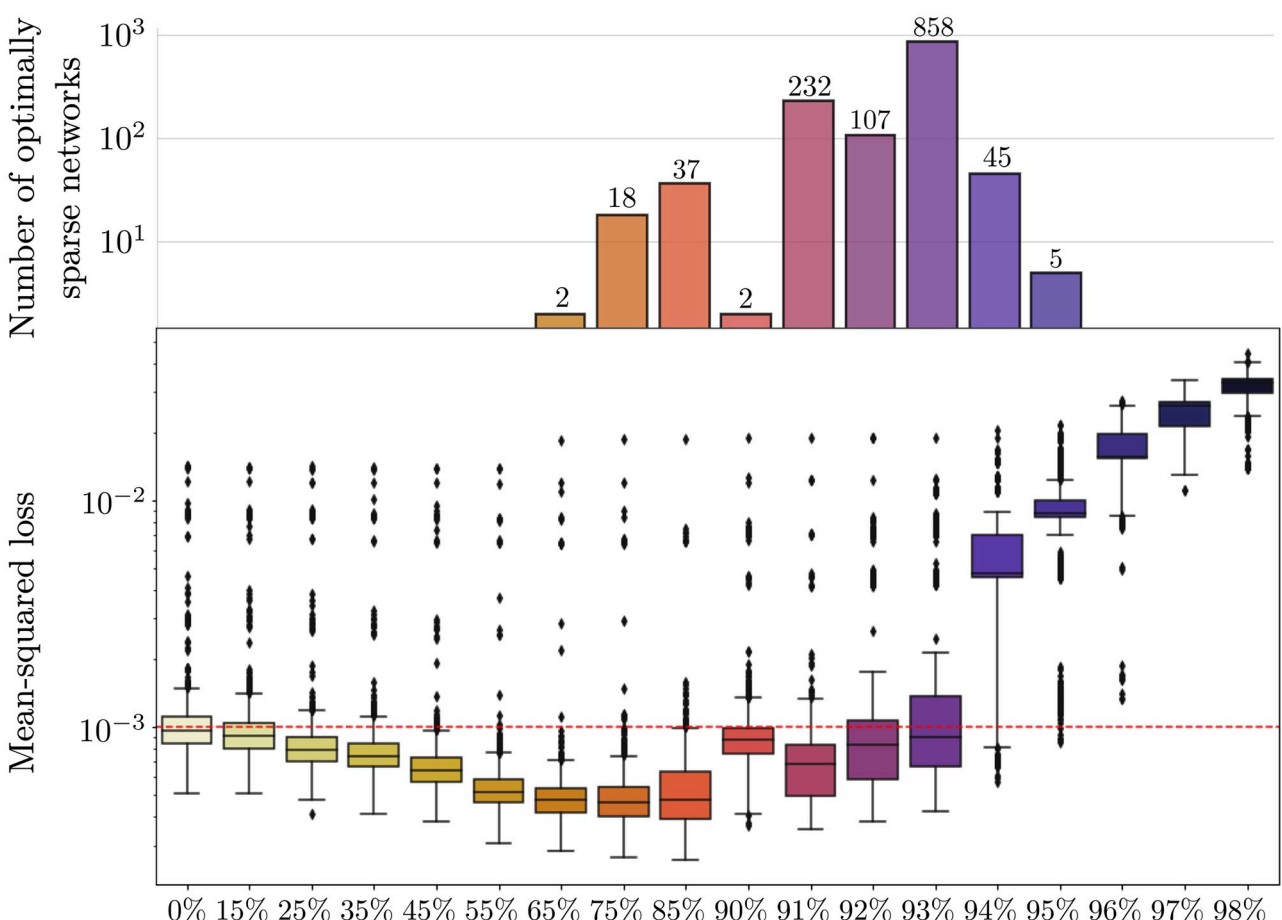

**Fig 4. Monte Carlo analysis of pruned networks.** 1320 networks are sequentially pruned and loss of the pruned networks at each sparsity percentage is recorded in the box plot. The bar plot records the number of networks that make it to the corresponding sparsity percentage before exceeding the hypothetical threshold ($10^{-3}$).

mass could rotate with angles $(\theta, \phi)$ in the *x-y* plane. See Fig 1 for more description of the simulated insect, S1 Table for the global model parameters, and S2 Table for the calculated model variables.

The computational model of the moth had three control variables and four state-space variables (as well as the respective state-space derivatives). This model is by definition

**Table 1. Number of remaining parameters in networks pruned to 93% sparsity.** This table gives the average number of remaining weights in each layer of the networks pruned to 93% sparsity. The variance on the number of connections, as well as the fraction of remaining connections are also given.

| Layer *i* | Average number remaining | Variance | Percentage remaining |
|---|---|---|---|
| 1 | 280 | 23 | 0.07 |
| 2 | 11199 | 0 | 0.07 |
| 3 | 11199 | 0 | 0.07 |
| 4 | 447 | 0.04 | 0.07 |
| 5 | 8 | 6 | 0.07 |

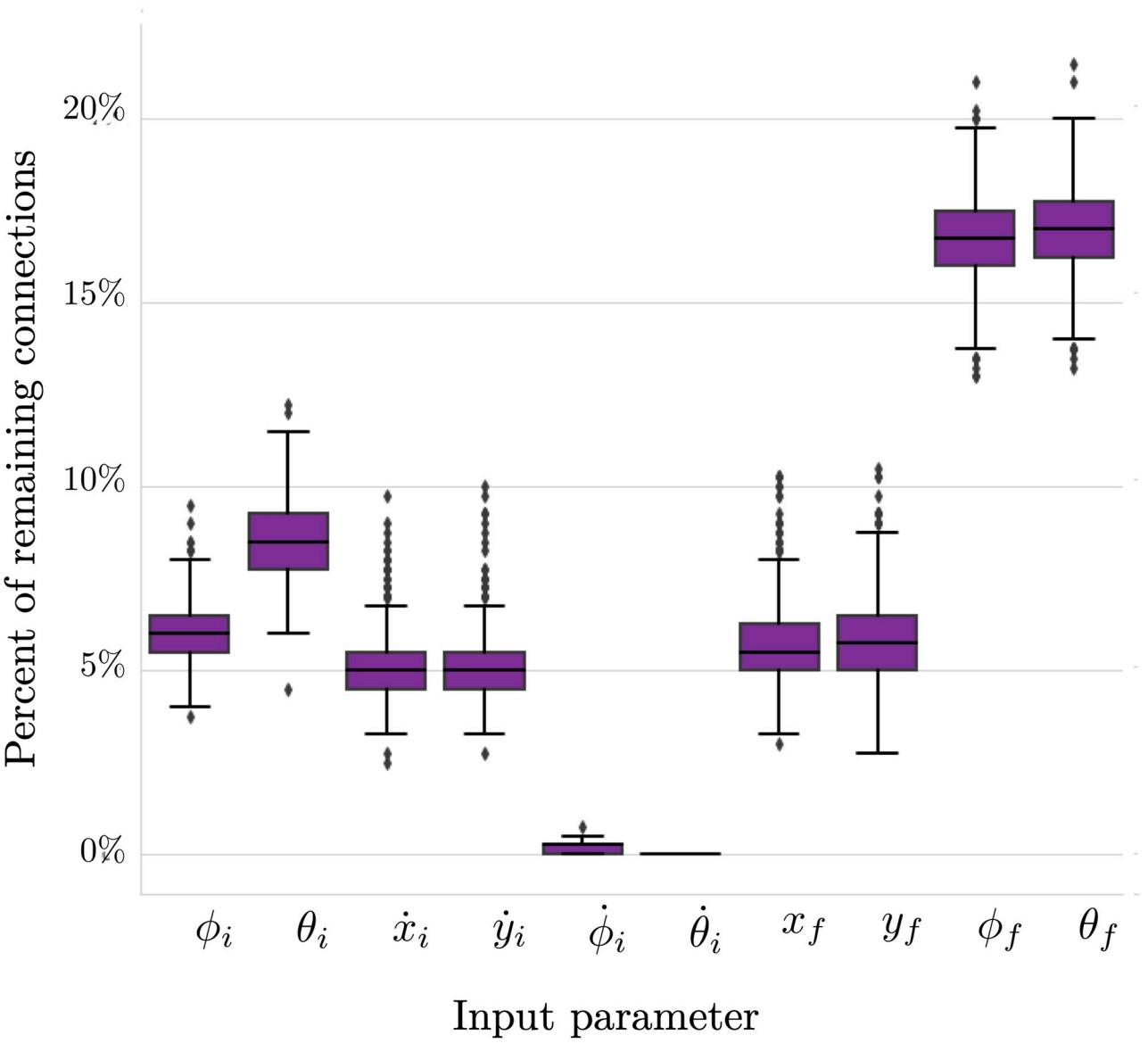

**Fig 5. Sparsity of input layer of networks pruned to 93% sparsity.** Each box represents the number of connections remaining between a parameter in the input layer and the first hidden layer. For all 858 networks in this group, $\dot{\theta}_i$ was pruned entirely from the network.

underactuated because the number of control variables is less than the degrees of freedom. The controls are as follows: $F$, the average force applied by the wings during each downstroke and upstroke; $\alpha$ the direction of force applied (with respect to the midline of the head-thorax mass); and $\tau$, the abdominal torque exerted about the pin joint connecting the two body segment masses (with its equal and opposite response torque). In addition to the downstroke or upstroke averaged forces, the model includes gravitational forces, abdominal torques, and drag forces on the body. The controls are randomized every 20 ms, which is approximately the period of the wing downstroke or upstroke for *M. sexta* (25 Hz wing beat frequency) [35]. Thus our model is basically a simplified two-body dynamical system with thrust vectoring. Since our dominant focus is on hovering flight, this provides a reasonable basis for examining

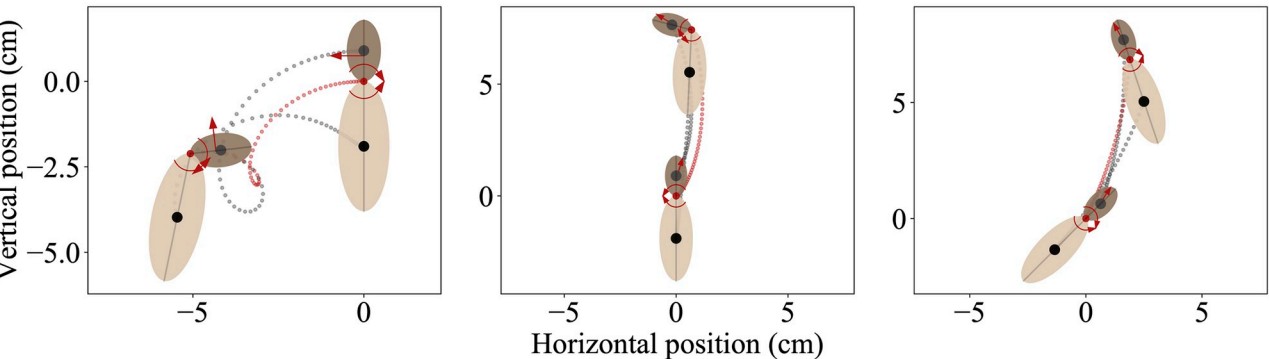

**Fig 6. Example trajectories of the simulated insects.** Each trajectory is 20 ms, and each starts at (x,y) = (0,0). Force ($F$) is indicated with the straight red arrow, and torque ($\tau$) is shown with the curved arrows at the thorax-abdomen joint (red dot). The center of mass of each body segment is shown with black dots.

the control consequences of pruning a deep neural network. Fig 6 shows three example hovering trajectories of the simulated insect. All trajectories begin at the origin ($(x, y) = (0, 0)$). The grey dotted lines show the trajectory of the center of mass of each body segment and the red dotted line shows the trajectory of the thorax-abdomen joint.

The motion of the moth state-space is described by four parameters ($x$: horizontal position, $y$: vertical position, $\theta$: head-thorax angle, and $\phi$: abdomen angle), as well as the respective state-space derivatives ($\dot{x}$: horizontal velocity, $\dot{y}$: vertical velocity, $\dot{\theta}$: head-thorax angular velocity, and $\dot{\phi}$: abdomen angular velocity). The x and y position indicate the position of the pin joint where the head-thorax connects with the abdomen.

## Generating training data

We used the ordinary differential equations from [31] (See Appendix, Equations 30–33) to generate a dataset for training the deep neural network. All simulated trajectories were started from the origin (*i.e.*, $(x_0, y_0) = (0,0)$). We randomly sampled initial horizontal velocity ($\dot{x}_0$), vertical velocity ($\dot{y}_0$), head-thorax angle ($\theta_0$), abdomen angle ($\phi_0$), head-thorax angular velocity ($\dot{\theta}_0$), and abdomen angular velocity ($\dot{\phi}_0$) as shown in S3 Table. We also randomly sampled force ($F$), force angle ($\alpha$), and torque ($\tau$) as shown in S3 Table. The training dataset is comprised of 10 million simulated trajectories and the test set contains an additional 5 million trajectories. The trajectories were simulated using the Python (Python Software Foundation, https://www.python.org/) function, `scipy.integrate.odeint` [36]. Fig 1 shows which variables were inputs and outputs from the differential equation solver.

## Data preparation for deep neural network training

The force ($F$) and force angle ($\alpha$) were converted to horizontal and vertical components ($F_x$ and $F_y$), using the following equations: $F_x = F \cdot cos(\alpha)$ and $F_y = F \cdot sin(\alpha)$. The data were split into training and validation sets for cross validation (80:20 split). The validation data is a sample used to provide an unbiased evaluation of a model fit while tuning the hyper parameters (such as number of hidden units, number of layers, optimizer, *etc.*). The data were scaled using a min-max scaler according to the training dataset and transformed values to be between −0.5 and + 0.5. The same scaler was then used to transform the validation and test data.

## Training and pruning a deep neural network

The deep, fully-connected neural network was constructed with ten input variables and seven output variables (see Fig 1). The initial and final state space conditions are the inputs to the network: $(\dot{x}_i,\ \dot{y}_i,\ \phi_i,\ \theta_i,\ \dot{\phi}_i,\ \dot{\theta}_i,\ x_f,\ y_f,\ \phi_f,\ \theta_f)$. The network predicts the control variables and the final derivatives of the state space in its output layer $(F_x,\ F_y,\ \tau,\ \dot{x}_f,\ \dot{y}_f,\ \dot{\phi}_f\ \dot{\theta}_f)$. The final derivatives of the state space were made outputs to be able to chain together 20 ms solutions to allow the moth to complete a complex trajectory for use in future work. The training and pruning protocols were developed using Keras [37] with the TensorFlow backend [32]. To scale up training for the statistical analysis of many networks, the training and pruning protocols were parallelized using the Jax framework [38].

To demonstrate the effects of pruning, the network was chosen to have a deep, feed-forward architecture with wide hidden layers (many more nodes than in the input and output layer). The network had four hidden layers with 400, 400, 400, and 16 nodes, respectively. Wide hidden layers were used rather than using a bottleneck structure (narrower hidden layer width) to allow the network to find the optimal mapping with little constraint, however, the specific choices of layer widths were arbitrary. The inverse tangent activation function was used for all hidden layers to introduce nonlinearity in the model. To account for the multiple outputs, the loss function was the uniformly-weighted average of the mean squared error for all the outputs combined.

$$MSE = \frac{1}{m}\sum_{i=1}^{m}(y_i - \hat{y}_i)^2 \tag{1}$$

For optimizing performance, there are several hyper-parameter differences in the TensorFlow model and the Jax model. In developing the training and pruning protocol in TensorFlow, the network was trained using the rmsprop optimizer with a batch size of $2^{12}$ samples. However, to scale up and speed up training we used the Jax framework, the Adam optimizer [39], and the batch size was reduced to 128 samples. Regularization techniques such as weight regularization, batch normalization, or dropout were not used. However, early stopping (with a minimum delta of 0.01 with a patience of 1000 batches) was used to reduce overfitting by monitoring the mean squared error.

After the fully-connected network is trained to a minimum error, we used the method of neural network pruning to promote sparsity between the network layers. In this work, a target sparsity (percentage of pruned network weights) is specified and those weights are forced to zero. The network is then retrained until a minimum error is reached. This process is repeated until most of the weights have been pruned from the network.

We developed two methods to prune the neural network: 1) a manual method that involves setting a number of weights to zero after each training epoch and 2) a method using TensorFlow's Model Optimization Toolkit [32] which involves creating a masking layer to control sparsity in the network. Both methods are described in detail in the following sections.

**Manual pruning.**   Algorithm 1 describes the a method of pruning in which the $n$ weights whose magnitudes are closest to zero are manually set to zero. If $N$ is the total number of weights in the network, the $n$ weights are chosen such that $n/N$ is equivalent to a specified pruning percentage (e.g. 15%, 25%, ..., 98%). After the $n$ weights are set to zero, the network is retrained for one epoch. This process is repeated until the loss is minimized. After the network has been trained to a minimum loss, we select the next pruning percentage from the predetermined list and repeat the retraining process. The entire pruning process is repeated until the network has been pruned to the final pruning percentage in the list.

**Algorithm 1:** Sequential pruning and fine-tuning

```
Train fully-connected model until loss is minimized
Define list of sparsity percentages
for Each sparsity percentage do
  while Loss is not minimized do
    for Each epoch do
      Set n weights to zero s.t. n/N equals the sparsity percentage;
      Evaluate loss;
      Update weights;
    end
  end
end
```

Upon retraining, the weights are able to regain a non-zero weight and the network is evaluated using these non-zero weights. Although this likely still captures the effects of pruning the network over the full training time, it is not true pruning in the sense that connections that have been pruned can regain weight.

**Pruning using Model Optimization Toolkit.**   The manual pruning method described above has the downside of allowing weights to regain non-zero value after training. These weights are subsequently set back to zero on the next epoch, but the algorithm does not guarantee that the same weights will be pruned every time.

To ensure weights remain pruned during retraining, we implemented the pruning functionality of a TensorFlow built toolkit called the Model Optimization Toolkit [32]. The toolkit contains functions for pruning deep neural networks. In the Model Optimization Toolkit, pruning is achieved through the use of binary masking layers that are multiplied element-wise to each weight matrix in the network. A four-layer neural network can be mathematically described the following way.

$$\hat{y} = \sigma_4(\mathbf{A_4} \ldots (\sigma_1(\mathbf{A_1}x))) \tag{2}$$

In Eq 2, the inputs to the network are represented by $x$, the predictions by $\hat{y}$, the weight matrices by $\mathbf{A_i}$, and the activation function by $\sigma_i$, where $i = 1, 2, 3, 4$ for the four layers of the network. During pruning, the binary masking matrix, $\mathbf{M_i}$ is placed between each layer.

$$\hat{y} = \mathbf{M_4} \circ \sigma_4(\mathbf{A_4} \ldots (\mathbf{M_1} \circ (\sigma_1(\mathbf{A_1}x)))) \tag{3}$$

In Eq 3, the binary masking matrices, $\mathbf{M_i}$, are multiplied element-wise to the weight matrices ($\circ$ denotes the element-wise Hadamard product). The sparsity of each layer is controlled by a separate masking matrices to allow for different levels of sparsity in each layer. Before pruning, all elements of $\mathbf{M_i}$ are set to 1. At each pruning percentage (e.g. 15%, 25%, . . ., 98%), the $n$ weights whose magnitudes are nearest to zero are found and the corresponding elements of the the $\mathbf{M_i}$ are set to zero. The network is then retrained until a minimum error is achieved. The masking layers are non-trainable, meaning they will not be updated during backpropagation. Then, the next pruning percentage is selected and the process is repeated until the network has been pruned to the final pruning percentage.

In the TensorFlow Model Optimization Toolkit, the binary masking layer is added by wrapping each layer into a prunable layer. The binary masking layer controls the sparsity of the layer by setting terms in the matrix equal to either zero or one. The masking layer is bi-directional, meaning it masks the weights in both the forward pass and backpropagation step, ensuring no pruned weights are updated [40]. Algorithm 1 shows the pruning paradigm utilizing the Model Optimization Toolkit.

**Algorithm 2:** Sequential pruning with masks and fine-tuning

```
Train fully-connected model until loss is minimized
```

```
Define list of sparsity percentages;
for Each sparsity percentage do
  Define pruning schedule using ConstantSparsity;
  Create prunable model by calling prune_low_magnitude;
  Train pruned model until loss is minimized;
end
```

Rather than controlling for sparsity at each epoch of training, as was done in the manual pruning method described above, we control for sparsity each time we want to prune more weights from the network. Sparsity is kept constant throughout each pruning cycle and therefore we can use TensorFlow's built-in functions for training the network and regularization.

## Preparing for statistical analysis of pruned networks

To be able to train and analyze many neural networks, the training and pruning protocols were parallelized in the Jax framework [38]. Rather than requiring data to be in the form of tensors (such as in TensorFlow), Jax is capable of performing transformations on NumPy [41] structures. Jax however does not come with a toolkit for pruning, therefore pruning by way of the binary masking matrices was coded into the training loop.

The networks were trained and pruned using a NVIDIA Titan Xp GPU operating with CUDA [42]. At most, 400 networks were trained at the same time and the total number of networks used in the analysis was 1320. These networks were all trained with identical architectures, pruning percentages, and hyper-parameters. The only difference between the networks is the random initialization of the weights before training and pruning. The Adam optimizer [39] and a batch size of 128 were used to speed up training and cross-validation was omitted. However, early stopping was used on the training data to avoid training beyond when the loss was adequately minimized. Additionally, early stopping was used to evaluate the decrease in loss across batches, rather than epochs.

## Discussion

In this study, we set out to investigate whether a sparse DNN can control a biologically relevant motor-task—in our case the dynamic control of a hovering insect. Taking inspiration from synaptic pruning found across wide ranging animal taxa, we pruned a DNN to different levels of sparsity in order to find the optimal sparse network capable of controlling moth hovering. The DNN uses data generated by the inertial dynamics model in [31] which models the forward problem of flight control. In this work, the DNN models the inverse problem of flight control by learning the controls given the initial and final state space variables.

Through this work, we found that sparse DNNs are capable of solving the inverse problem of flight control, i.e. predicting the controls that are required for a moth to hover to a specified state space. In addition, we demonstrate that across many networks, a network can be pruned by as much as 93% and perform comparably to the median performance of a fully-connected network. However, there are sharp performance limits and most networks pruned beyond 93% see a breakdown in performance. We found that although uniform pruning was not enforced, on average, each layer in the network pruned to match the overall sparsity (i.e. sparsity of each layer was 93% for networks pruned to overall sparsity of 93%). Finally, we looked at the sparsity of individual layers and found that the initial head-thorax angular velocity is consistently pruned from the input layer of networks pruned to 93% sparsity, indicating a redundancy in the forward original model.

Though we have shown that a DNN is capable of learning the controls for a flight task, there are several limitations to this work. Firstly, though the model in [31] used to generate the training data provided control predictions for accurate motion tracking in a two-dimensional

task, biological reality is more rich and complex than can be captured by the forward model. Thus, since the DNN is trained with this data, it is only capable of learning the dynamics captured in the model in [31]. Furthermore, the size, shape, and body biomechanics of this systems all matter. This study uses the same global parameters across the data set (see S1 Table), but, in reality, these parameters vary significantly (across insect taxa and within the life of an individual) and this likely affects the performance of the DNN.

We have shown here that DNNs are capable of learning the inverse problem of flight control. The fully-connected DNN used here learned a nonlinear mapping between input and output variables, where the inputs are the initial and final state space variables and the outputs are the controls and final velocities. A fully-connected network can learn this task with a median loss of 7.9 x $10^{-4}$. However, due to the random initialization of weights preceding training, some networks perform as much as an order of magnitude worse (see Fig 4). This suggests that the performance of a trained DNN is heavily influenced by the random initialization of its weights.

We used magnitude-based pruning to sparsify the DNNs in order to find the optimal, sparse network capable of controlling moth hovering. For the task of moth hovering, a DNN can be pruned to approximately 7% of its original network weights and still perform comparably to the fully-connected network. The results of this analysis show that when trained to perform a biological task, fully-connected DNNs are indeed over-parameterized. Much like their biological counterparts, DNNs do not require fully-connected connectivity to accomplish this task. Additionally, flying insects maneuver through high noise environments and therefore perfect flight control is traded for robustness to noise. It is therefore assumed that the data has a given signal-to-noise ratio or performance threshold. The performance threshold represented by the red dashed line in Figs 2, 3 and 4 was arbitrarily chosen to represent a loss comparable to the loss of the fully-connected network (i.e. 0.001). In other words, this line represents a noise threshold, below which the network is considered well-performing and adapted to noise. It has been shown that biological motor control systems are adapted to handle noise [43]. Biological pruning may be a mechanism for identifying sparse connectivity patterns that allow for control within a noise threshold.

On average, when the networks are pruned beyond 7% connectivity, there is a dramatic performance breakdown. Beyond 93% sparsity, the performance of the networks break down because too many critical weights have been removed. A significant proportion of the 1320 networks breakdown before they reach 7% connectivity (approximately 30% of the networks). This again supports the aforementioned claim that the random initialization of the weights before training affects the performance of a DNN and can be exacerbated by neural network pruning. Additionally, this shows that there exists a diversity of network structures that perform within the bounds of the noise threshold.

To investigate the substructure of the well-performing, sparse networks, we looked closer at the subset of networks that were optimally sparse at 93% pruned (858 networks). We have shown that the average sparsity of each layer in this subset is uniform, meaning each of the five layers have approximately 7% of their original connections remaining. However, the variance on the number of remaining connections between input layer and first hidden layer and between the final hidden layer and the output layer is markedly higher than the variance in the weight matrices between the hidden layers. This suggests that in networks pruned to 93% sparsity, the greatest amount of change in network connectivity occurs in input and output layers. However, there are notable features in the connectivity between the input and first hidden layer that are consistent across the 858 networks. Fig 5 shows that the input parameter, initial head-thorax angular velocity ($\dot{\theta}_i$), is completely pruned from all of the 858 networks. The

initial abdomen angular velocity ($\dot{\phi}_i$) is also almost entirely pruned from all of the networks. All of the other input parameters maintain an median of at least 5% connectivity to the first hidden layer. The complete pruning of $\dot{\theta}_i$ suggests a redundancy in the original forward model. However, this redundancy makes physical sense because $\theta_i$ and $\phi_i$ are coupled in the original forward model.

The results of this study pose an interesting question about how the size of the initial network architecture affects the resultant pruning statistics. The networks pruned in this study are feed-forward, each with four hidden layers with 400, 400, 400, 16 nodes, respectively. This choice of architecture is somewhat arbitrary, however through the process of tuning and cross-validating the fully-connected network, we converged to a set of hyperparameters (including the size of the hidden layers) which resulted in the most optimally performing network. To begin to explore the effect that initial network architecture size has on the pruning statistics, we repeated the experiment with increasingly smaller network architectures. For example, in S1 Fig we trained 400 networks, each with four hidden layers with 200, 200, 200, 8 nodes. S2 Fig shows the same results for networks of sizes 100, 100, 100, 8 and S3 Fig shows the results for networks of size 50, 50, 50, 8. These decreases in hidden layer widths correspond to a decrease in the total number of weights across the networks from 330, 512 (for the original networks in Fig 4) to 83, 656 (S1 Fig), 21, 856 (S2 Fig), and 5, 956 (S3 Fig). Across all networks, there is a slight improvement in performance for low levels of pruning. All networks show a performance breakdown, however the sparsity at which the breakdown occurs changes with the size of the network. For example, the networks in S3 Fig show a performance breakdown at 65% or when there are 2, 084 weights remaining. This is compared to the original 1320 networks which showed a performance breakdown at 93% or when there are 23, 135 weights remaining. The initial architecture of the network affects the achievable sparsity by the pruning protocol employed here. Additionally, smaller network architectures result in more volatility when higher levels of sparsity are reached. However, the results of these preliminary experiments only begin to explore the relationship between initial network architecture and resultant pruning statistics. We found that as the network architecture is made smaller, the raw number of parameters post-pruning is fewer. Whether these extra small networks are as robust to noise or better at generalizing to unseen data is yet to be seen. It is also unclear what the optimal starting architecture size should be because large, over-parameterized networks are thought to be more efficient to optimize via gradient descent [44]. As stated, these preliminary experiments open up many interesting questions to be explored in future work.

In this work, we have shown that a sparse neural network can learn the controls for a biological motor task and we have also shown, via Monte Carlo simulations, that there exists at least some aspects of network structure that are stereotypical. There are several computationally non-trivial extensions to the work presented here. Firstly, network analysis techniques (such as network motif theory) could be used to further compare the pruned networks and investigate the impacts of neural network structure on a control task. Network motifs are statistically significant substructures in a network and have been shown to be indicative of network functionality in control systems [45]. Other areas of future work include investigating the sparse network's response to noise and changes in the biological parameters. Biological control systems are adapted to function adequately in the presence of noise. Pruning improves the performance of neural networks up to a certain level of sparsity, however the effects of noise on this bio-inspired control task are yet to be explored. Furthermore, the size and shape of a real moth can change rapidly (e.g. change of mass after feeding). The question of whether sparsity improves robustness in the face of such physical parameters could also be a future extension of this work.

## Conclusion

Synaptic pruning has been shown to play a major role in the refinement of neural connections, leading to more effective motor task control. Taking inspiration from synaptic pruning in biological systems, we apply the equally thoroughly investigated method of DNN pruning to the inverse problem of insect flight control. We use the inertial dynamics model in [31] to simulate examples of *M. sexta* hovering flight. This data is used to train a DNN to learn the controls for moving the simulated insect between two points in state-space. We then prune the DNN weights to find the optimally sparse network for completing flight tasks. We developed two paradigms for pruning: via manual weight removal and via binary masking layers. Furthermore, we pruned the DNN sequentially with retraining occurring between prunes. Monte Carlo simulations were also used to quantify the statistical distribution of network weights during pruning to find similarities in the internal structure across pruned networks. In this work, we have shown that sparse DNNs are capable of predicting the controls required for a simulated hawkmoth to move from one state-space to another.

## Supporting information

**S1 Fig. Pruning results for networks (layer widths: 200, 200, 200, 8.** 400 networks, each with four hidden layers with 200, 200, 200, and 8 nodes, respectively, are sequentially pruned and loss of the pruned networks at each sparsity percentage is recorded in the box plot. The bar plot records the number of networks that make it to the corresponding sparsity percentage before exceeding the hypothetical threshold ($10^{-3}$).
(PDF)

**S2 Fig. Pruning results for networks (layer widths: 100, 100, 100, 8.** 400 networks, each with four hidden layers with 100, 100, 100, and 8 nodes, respectively, are sequentially pruned and loss of the pruned networks at each sparsity percentage is recorded in the box plot. The bar plot records the number of networks that make it to the corresponding sparsity percentage before exceeding the hypothetical threshold ($10^{-3}$).
(PDF)

**S3 Fig. Pruning results for networks (layer widths: 50, 50, 50, 8.** 400 networks, each with four hidden layers with 50, 50, 50, and 8 nodes, respectively, are sequentially pruned and loss of the pruned networks at each sparsity percentage is recorded in the box plot. The bar plot records the number of networks that make it to the corresponding sparsity percentage before exceeding the hypothetical threshold ($10^{-3}$).
(PDF)

**S4 Fig. Error evaluation (before pruning).** Error evaluation of a fully-connected network before any pruning. The seven parameters shown are the outputs of the network, the three control variables and the final derivatives of the state space. The residual plots are also shown (denoted by Actual—Prediction).
(PDF)

**S5 Fig. Error evaluation (with pruning).** See S4 Fig. Note that axes for residual plots are scaled to include the max outliers.
(PDF)

**S1 Table. Global parameters for the moth model.** A table of all the required parameters to recreate the simulated model of a moth.
(PDF)

**S2 Table. Calculated variables for moth model.** A table of all the required calculated variables to recreate the simulated model of a moth.
(PDF)

**S3 Table. Initial state space and controls for generating training data.** The following variables are the required state space and control variables along with their ranges for random uniform sampling.
(PDF)

**S4 Table. Final state space variables.** The following variables are the output from the differential equation solver. These variables describe the final state space of the insect after 20ms.
(PDF)

## Acknowledgments

We gratefully acknowledge the support of NVIDIA Corporation with the donation of the Titan Xp GPU used for this research and Henning Lange for his valuable help in developing the JAX code.

## Author Contributions

**Conceptualization:** Olivia Zahn, Callin Switzer, Thomas L. Daniel, J. Nathan Kutz.

**Data curation:** Olivia Zahn, Jorge Bustamante, Jr., Callin Switzer.

**Formal analysis:** Olivia Zahn, Jorge Bustamante, Jr., Callin Switzer.

**Funding acquisition:** Thomas L. Daniel, J. Nathan Kutz.

**Investigation:** Olivia Zahn, Jorge Bustamante, Jr., Callin Switzer.

**Methodology:** Olivia Zahn, Callin Switzer.

**Project administration:** Thomas L. Daniel, J. Nathan Kutz.

**Software:** Jorge Bustamante, Jr.

**Supervision:** Thomas L. Daniel, J. Nathan Kutz.

**Validation:** Olivia Zahn.

**Visualization:** Olivia Zahn, Callin Switzer.

**Writing – original draft:** Olivia Zahn, Jorge Bustamante, Jr., Callin Switzer, Thomas L. Daniel, J. Nathan Kutz.

**Writing – review & editing:** Olivia Zahn, Jorge Bustamante, Jr., Thomas L. Daniel, J. Nathan Kutz.

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
