## [Decision Letter · Decision Letter 0]

13 Apr 2022

Dear Thomas,

Thank you very much for submitting your manuscript "Pruning deep neural networks generates a sparse, bio-inspired nonlinear controller for insect flight" for consideration at PLOS Computational Biology.

As with all papers reviewed by the journal, your manuscript was reviewed by members of the editorial board and by several independent reviewers. In light of the reviews (below this email), we would like to invite the resubmission of a significantly-revised version that takes into account the reviewers' comments.

The Bustamante model is unavailable online for reference. This appears to be a model for the control of abdominal posture during flight. The actual flight mechanics are controlled by a combination of indirect and direct flight muscles operating the wings (Balint & Dickinson, 2001). How do the investigators reconcile these two models?

There is an extensive literature on the synaptic networks underlying insect flight. Pearson & Robertson, 1987; Robertson & Pearson, 1983, 1985; Robertson & Reye, 1988, Burrows, 1996. Do any of these models exhibit the backpropagation network motif, characteristic of machine learning and deep learning.

How are off-target contacts withdrawn? Reference 5 does not contain the term pruning. There appear to be two existing proposed mechanisms for “synaptic pruning”. Synaptic Refinement involves the growth cone (Vonhoff & Keshishian,2017) The other mechanism is phagocytic glial cells (Schafer & Stevens, 2013. Which mechanism is being modeled here?

We cannot make any decision about publication until we have seen the revised manuscript and your response to the reviewers' comments. Your revised manuscript is also likely to be sent to reviewers for further evaluation.

Sincerely,

Joseph Ayers, PhD

Associate Editor

PLOS Computational Biology

Daniele Marinazzo

Deputy Editor

PLOS Computational Biology

The Bustamante model is unavailable online for reference. This appears to be a model for the control of abdominal posture during flight. The actual flight mechanics are controlled by a combination of indirect and direct flight muscles operating the wings (Balint & Dickinson, 2001). How do the investigators reconcile these two models?

There is an extensive literature on the synaptic networks underlying insect flight. Pearson & Robertson, 1987; Robertson & Pearson, 1983, 1985; Robertson & Reye, 1988, Burrows, 1996. Do any of these models exhibit the backpropagation network motif, characteristic of machine learning and deep learning.

How are off-target contacts withdrawn? Reference 5 does not contain the term pruning. There appear to be two existing proposed mechanisms for “synaptic pruning”. Synaptic Refinement involves the growth cone (Vonhoff & Keshishian,2017) The other mechanism is phagocytic glial cells (Schafer & Stevens, 2013. Which mechanism is being modeled here?

Reviewer's Responses to Questions

**Comments to the Authors:**

Reviewer #1: I found this an interesting paper, and the central concept of using network pruning to improve the performance of a model insect flight controller certainly seems worthwhile. Although I am familiar with the theory of deep neural networks, I am not a practitioner myself, so have confined most of my comments below to the aspects related to insect flight dynamics and control. It will be important to ensure that at least one of the other reviews secured is from a reviewer with specific expertise in deep networks.

Flight dynamics model:

1. The description of the flight dynamics model relies heavily on an unpublished manuscript cited as being in revision at another journal. For example, the authors write (p8): “We used the ordinary differential equations from [27] (See Appendix, Equations 30-33) to generate a dataset for training the deep neural network.” As the present submission has no appendix, I assume that this refers to the appendix of Ref. 27. The summary of the model in Fig. 1 treats these ODEs as a black box, and as they are not stated elsewhere in the manuscript, it follows that the work is not described sufficiently well in the manuscript to be reproducible in its current form. I did try reviewing the supporting GitHub briefly to look for the equations, but don’t feel that I should have to go to this trouble or need to read Python to understand the modelling.

2. Taking the model at face value, some further justification and explanation is necessary to demonstrate that it really does provide a meaningful model of insect flight dynamics. Although the limitations of the model are mentioned in general terms on p12, there are several specific features of the model that seem surprising and that would therefore benefit from further justification and explanation.

a. The dynamics of flight result from gravitational and aerodynamic forces, but the summary in Fig. 1 only shows one external force “F”. Gravitational acceleration is listed as a model parameter in Table 2, so I assume that “F” refers to the aerodynamic force only, and that the insect’s body weight is included separately in the model. (If this were not so, such that “F” was in fact the total net force, then I would have other concerns on the constraints imposed by the modelling.) Either way, the fact that there is any uncertainty on this point at all reinforces the need to improve a full explanation of the model, to avoid the reader having to trawl through code to find out. Providing the ODEs is also important to understanding why the network inputs and outputs are set up as they are. Besides providing all relevant equations, I would strongly suggest showing all the relevant forces and torques in Fig. 1 for clarity.

b. Under the model, the force F can act in any direction with respect to the axes of the head-thorax element, as its angle alpha is drawn from a uniform distribution on the interval (0, 2pi). This seems implausible, and leaves me feeling that the flight dynamics model is better viewed as a model of a two-body system with thrust vectoring, rather than as a model of insect flight per se. Related to this, it seems from the parameterization that the force F is unaffected by the insect’s body motion, which runs contrary to the results or assumptions of most other insect flight dynamics modelling.

c. This being so, it would be reassuring to see some validation of the model. Again, I presume that this is provided by Ref. 27, but as things stand the reader is left without either a principled derivation or an empirical validation to go on. On this basis, although it is clear that the authors have constructed a pruned network architecture that is capable of controlling the motor system that they have modelled, it is less clear what this has to do with insect flight.

I trust that these comments will be reasonably straightforward to address, but would like to see some more detailed explanation, justification, and validation of the flight dynamics model before making any final recommendation.

Network pruning:

The authors conclude (p13) that “For the task of moth hovering, a DNN can be pruned to approximately 7% of its original network weights and still perform comparably to the fully-connected network.” Although the dimensions of the input and output layers are uniquely defined by the model of moth hovering that the authors have used, it appears that the authors have made an arbitrary choice to include 400 nodes in three hidden layers, and 16 nodes in another. I would assume that the quantitative conclusions on pruning rest heavily on this choice, so would like to see some discussion or exploration of this point.

Reviewer #2: This is a fascinating study by Olivia Zahn et al which uses deep learning neuronal networks (DNN) that model the hovering flight of moths (Manduca sexta) in order to explore how pruning of a fully connected network affects the performance of the network model. The authors do a great job in introducing and discussing the role of synaptic pruning in the development of neuronal networks as has been shown in many different biological systems such as the classic Hubel Wiesel studies of the visual cortex. This broad discussion makes this study interesting for a general readership.

In their study the authors train a fully connected network to generate hovering with a minimum error. Subsequently they used manual and computational approaches to prune these fully connected DNNs to different levels of sparsity to find the optimal sparse network configuration, which is still capable of controlling moth hovering. Perhaps, unexpected to the authors (but not unexpected to this reviewer) the authors find that their networks perform better than the median performance of the fully connected networks when pruned up to 85% sparsity. Indeed, of the 1320 DNNs that they trained 858 were optimally sparse at 93% sparsity. It was also surprising that some parameters, such as the initial head-thorax angular velocity, were completely pruned out as it had no impact on the output and predictive power of the network. The authors also found that these pruned networks exhibit sharp performance limits when pruned beyond 93%. This was found at all layers of the networks.

Of course, one major caveat is that the biological reality is more rich, and complex than the forward model networks that were generated to simulate moth hovering. But, to this reviewer, this is a very minor issue that does not detract from the fundamental lessons learned from this study: i.e. that sparsity imbues neuronal network with increased capabilities. Many popular, contemporary, computational models of rhythmogenic networks are fully connected ball-and-stick models which are inspired e.g. by the very small network of the stomatogastric ganglion. However, in reality, fully connected networks are an exception and most networks are not fully connected but exceedingly sparse. With the sparsity comes e.g. a characteristic cycle-by-cycle variability that is often ignored. The authors have done a fabulous job in discussing the universal role of pruning in their discussion, but I highly recommend that the authors also discuss the existence of sparsely connected networks in biology. This is currently missing in this manuscript. One such network is the mammalian respiratory network which is sparsely connected. This network has to be extremely flexible, yet also very robust. It is characterized by a high cycle variability, and sparse connectivity. The cortex is also sparsely connected. Indeed, because of the sparse connectivity of mammalian neuronal networks, so little is known about the circuit diagrams underlying mammalian neuronal networks: it is exceedingly difficult to reproducibly find connected pairs of neurons. Because of the sparsity no mammalian network circuit exists that looks like the STG or the leech heartbeat system. Thus, I highly recommend that the authors discuss the presence of sparsely connected network which will provide further validity for the fundamental conclusions drawn in this very inspiring computational study.

**Have the authors made all data and (if applicable) computational code underlying the findings in their manuscript fully available?**

Reviewer #1: Yes

Reviewer #2: Yes

PLOS authors have the option to publish the peer review history of their article (what does this mean?). If published, this will include your full peer review and any attached files.

Reviewer #1: No

Reviewer #2: No
---

## [Decision Letter · Decision Letter 1]

24 Aug 2022

Dear Thomas,

We are pleased to inform you that your manuscript 'Pruning deep neural networks generates a sparse, bio-inspired nonlinear controller for insect flight' has been provisionally accepted for publication in PLOS Computational Biology.

Best regards,

Joseph Ayers, PhD

Academic Editor

PLOS Computational Biology

Daniele Marinazzo

Section Editor

PLOS Computational Biology

Reviewer's Responses to Questions

**Comments to the Authors:**

Reviewer #1: The authors have addressed all of the points raised by the Editor and the reviewers. Their new analysis testing the effects of pruning networks of different sizes is especially useful (Figs. S1-S3), and could perhaps be brought forward from the Discussion to the Results. These supplementary analyses demonstrate the same qualitative principle as the main Results, viz. that substantial pruning of the network is always possible without substantial loss of performance. Quantitatively, its results are quite different, with an appreciable breakdown occurring beyond 65-75% sparsity, as compared to the 93% sparsity identified for the larger networks used in the main analysis. It would therefore be appropriate to qualify the quantitative statements that the authors make in their Discussion accordingly. In general, when stating that pruning to 7% of the original weights (or 93% sparsity) causes no appreciable loss of performance, etc. I would recommend adding the caveat "for networks of the dimensions tested here". For example, the authors' quantitative statement that "For the task of moth hovering, a DNN can be pruned to approximately 7% of its original network weights and still perform comparably to the fully-connected network." is true in the narrow sense of the network tested, but reads as being rather too sweeping a statement in light of Figs S1-3. None of this undermines the central conclusion that substantial pruning is possible; rather, my advice is not to place too much emphasis on the 7% (93%) figure. I congratulate the authors on their interesting, and indeed groundbreaking, study.

Reviewer #2: I have no further comments – the manuscript has greatly improved

**Have the authors made all data and (if applicable) computational code underlying the findings in their manuscript fully available?**

Reviewer #1: Yes

Reviewer #2: Yes

PLOS authors have the option to publish the peer review history of their article (what does this mean?). If published, this will include your full peer review and any attached files.

Reviewer #1: No

Reviewer #2: No

---

## [Editor Report · Acceptance letter]

22 Sep 2022

PCOMPBIOL-D-22-00064R1 

Pruning deep neural networks generates a sparse, bio-inspired nonlinear controller for insect flight

Dear Dr Zahn,

I am pleased to inform you that your manuscript has been formally accepted for publication in PLOS Computational Biology. Your manuscript is now with our production department and you will be notified of the publication date in due course.

With kind regards,

Agnes Pap
